# Screening of Sourdough Starter Strains and Improvements in the Quality of Whole Wheat Steamed Bread

**DOI:** 10.3390/molecules27113510

**Published:** 2022-05-30

**Authors:** Juan Shen, Kexin Shi, Hua Dong, Kesheng Yang, Zhaoxin Lu, Fengxia Lu, Pei Wang

**Affiliations:** 1College of Food Science and Technology, Nanjing Agricultural University, Nanjing 210095, China; 2016027@njau.edu.cn (J.S.); kekenvshi@163.com (K.S.); 2019108039@njau.edu.cn (K.Y.); fmb@njau.edu.cn (Z.L.); 2COFCO Donghai Grain and Oil Industry (Zhangjiagang) Co., Ltd., Suzhou 215632, China; donghua@cofco.com

**Keywords:** sourdough starter, whole wheat steamed bread, quality, volatile compounds

## Abstract

In this study, yeast, lactic acid bacteria, and acetic acid bacteria were isolated from traditional Chinese sourdough to enhance the organoleptic quality of whole wheat steamed bread. The *Saccharomyces cerevisiae*, *Lactobacillus johnsonii*, and *Acetobacter pasteurianum* showed superior fermentability and acid production capacity when compared with other strains from sourdough, which were mixed to produce the compound starter. It was found that the volume of whole wheat steamed bread leavened with compound starter increased by 12.8% when compared with that of the whole wheat steamed bread made by commercial dry yeast (DY-WB). A total of 38 volatile flavors were detected in the whole wheat steamed bread fermented by the compound starter (CS-WB), and the type of volatile flavors increased by 14 species when compared to the bread fermented by the dry yeast. In addition, some unique volatile flavor substances were detected in CS-WB, such as acetoin, 3-hydroxy-butanal, butyraldehyde, cuparene, etc. Moreover, the hardness and the chewiness of CS-WB decreased by 31.1 and 33.7% when compared with DY-WB, respectively, while the springiness increased by 10.8%. Overall, the formulated compound starter showed a desirable improvement in the whole wheat steamed bread and could be exploited as a new ingredient for steamed bread.

## 1. Introduction

Steamed bread is one of the traditional staple foods, accounting for over 40% of wheat products in China. It is usually prepared from wheat flour, microbial starter, and water. When compared to traditional sourdough, steamed bread made with commercial dry yeast has a plain flavor, poor aroma, and low overall sensory quality. However, when compared to refined wheat flour, whole wheat flour contains higher levels of vitamins, minerals, dietary fiber, antioxidants, and other phytochemicals such as carotenoids, flavonoids, and phenolic acids [1,2]. Studies have shown that the regular consumption of whole-grain foods can effectively reduce the risk of chronic diseases, such as obesity, cardiovascular and cerebrovascular diseases, and diabetes [3]. However, the presence of insoluble polysaccharides in bran and the physical structure of bran particles affect the formation of the gluten network and water distribution of dough, which further affect the texture of the product [4]. Therefore, how to produce high-quality whole wheat flour products is an urgent problem to be solved at this stage.

Sourdough is a traditional starter to produce fermented products made from flour. In Europe and other countries, sourdough is usually used to make baked bread, especially traditional sourdough bread and rye bread [5]. In China, sourdough is also called “Liaomian” and “Jiaozi”, which are mainly used to make traditional fermented wheat products such as steamed bread and buns [6,7]. Traditional sourdough involves a complex micro-ecological environment, mainly including yeasts, lactic acid bacteria, and acetic acid bacteria [8]. The sourdough with a multi-strain mixed fermentation system develops a unique flavor and nutritional quality [9]. To date, more than 70 kinds of lactic acid bacteria and 25 kinds of yeast have been identified in the sourdough system [6]. Both the strains have positive impact on the flavor, taste, and nutritional value of products [10]. The quality of traditional sourdough fermentation is unstable between batches, because the microbial flora is complex and susceptible to endogenous or exogenous factors [11]. Moreover, most of the traditional starters are naturally fermented, and they are produced by households in workshops. The sanitary conditions are poor, and they may contain some miscellaneous bacteria or even harmful bacteria. It is thus difficult to apply to industrial production. Therefore, the separation of dominant strains from Chinese traditional sourdough (CTS) has gradually attracted attention in recent years. Wu et al. [12] fermented steamed bread with different mixing ratios of lactic acid bacteria, and studied the effect on the specific volume, crumb texture, volatile profile, and other aspects of steamed bread quality. Fekri et al. [13] collected sourdough from traditional locations and selected excellent strains with phytate degrading ability to make whole wheat bread, and the results showed that the whole wheat bread prepared by *Kluyveromyces aestuarii* possessed the highest porosity percentage and the lowest hardness among all the tested strains [13]. In the steamed bread fermentation process, yeast and lactic acid bacteria played synergistical roles in giving the sourdough fermented product a unique texture and flavor [7], and the taste in the process is more mellow and delicate, as well as the nutritional value of the products being improved, especially in terms of protein quality and mineral utilization [14]. Since the sourdough fermentation technology has a positive effect on imparting a special flavor to steamed bread, improving its texture, as well as enhancing its nutrition, sourdough fermentation technology can be used to improve the organoleptic quality in whole wheat steamed bread.

Against this background, the aim of this study was to enhance the organoleptic quality of whole wheat steamed bread by the microbial strains isolated from CTS. The dominant strains were characterized from CTS and the optimal strains were selected based on fermentability and acid production capacity. The maximum benefits on the whole wheat steamed bread quality were realized by the compound starter with mixed strains. The results of this study could contribute to exploiting a new compound starter to enhance the quality of whole wheat steamed bread.

## 2. Results and Discussion

### 2.1. Acidity and Microbial Counts of CTS

As shown in Table 1, the pH values of CTS from different regions were between 3.61 and 5.19. The TTA value showed an opposite trend, differing from 6.40 mL to 16.32 mL. Acidity is an important index to measure the maturity of sourdough, and the sourdough maturity is decided by the organic acids produced by LAB metabolism. Different degrees of acidification can affect the formation components of sourdough structure, such as starch, gluten, and arabinoxylan [15]. According to previous studies, the sourdough acidity is usually between 4.0 and 4.5 [16]. The differences might result from changes in processing technology, such as different starter, flour type, and fermentation conditions [17].

According to the microbial counts, the yeast had the highest density (6.73 Log cfu/g) in CTS1, while the LAB had the highest density (9.57 Log cfu/g) in CTS4, among others. There were some differences in LAB numbers in different CTS, with a maximum difference of three to four orders of magnitude. The content and proportion of yeast and LAB were different in the samples. The difference in the counts of microbial levels reflected the quality difference of sourdough, which may be determined by many factors, such as the flour type, production technology, added ingredients, fermentation environment, storage time and conditions, etc. [18].

### 2.2. Isolation and Identification of LAB and Yeast in Sourdough

Observing the colony growth on YPD plate and MRS plate, we found 40 suspected strains of yeast and 40 suspected strains of LAB. A single bacterial colony was obtained by streaking pure culture; 15 strains of yeast and 20 strains of LAB were screened by colony morphology and electron microscope observation.

After comparing the sequence homology of dominant strains in the gene bank, those with a sequence homology higher than 95% were selected (Table 2) to build the phylogenetic trees (Figure 1 and Figure 2). Among the 15 strains of yeast and 20 strains of bacteria, three yeast species and six LAB species were identified: *Saccharomyces cerevisiae* (six strains), *Wicherhamomyces anomalus* (four strains), *Pichia kudriavzevii* (three strains), *Lactobacillus johnsonii* (five strains), *Pediococcus pentosaceus* (four strains), *Levilactobacillus brevis* (three strains), *Lacticaseibacillus casei* (two strains), *Lacticaseibacillus paracasei* (two strains), *Pediococcus acidilactici* (one strain), and *Acetobacter pasteurianus* (four strains). *Saccharomyces cerevisiae* is the dominant yeast in sourdough. *Wicherhamomyces anomalus* is also a common yeast found in sourdough, which can produce substances to inhibit fungus growth during dough fermentation and bread preservation [19,20]. It has been reported that steamed bread fermented by *Pichia kudriavzevii* has higher consumer recognition than ordinary steamed bread [21]. *Lactobacillus johnsonii* has been proven to reduce the proliferation of Clostridium perfringens in the intestinal tract of poultry, and with its own envelope forms, *Lactobacilluis johnsonii* plays an important role in inhibiting the proliferation of harmful bacteria [22]. *Levilactobacillus brevis* can perform heterotypic fermentation and produce lactic acid, alcohol, organic acids, aldehydes, etc., which are important sources of the sourdough fermentation flavor [23]. In addition, *Acetobacter* was isolated from samples in the Anhui province. The acetic acid produced by *Acetobacter* metabolism is the characteristic flavor component of sourdough and can inhibit the growth of spoilage microorganisms.

### 2.3. Selection of Starters

#### 2.3.1. Selection of Strains

Due to the fermentation capacity and acid production capacity, three yeast strains, six LAB strains, and one acetic acid bacteria strain were isolated in this study. The gas-producing abilities of different yeast strains are shown in Figure 3A. The fermentability of yeast strains was evaluated by the drainage rate in 0–4 h as the index of fermentability. The speed and gas production of yeast are comprehensively listed in Figure 3A. The low homology in the early identification of strains Y-4 and Y-11 were not included in this study. Finally, three strains with relatively good fermentation capacity were selected from 13 yeast strains for subsequent experiments, which were Y-2 (*Saccharomyces cerevisiae*), Y-6 (*Pichia kudriavzevii*), and Y-9 (*Wicherhamomyces anomalus*). *Saccharomyces cerevisiae* proved to be a superior starter due to its good gas production and low pH tolerance [24]. *Wickerhamomyces anomalus* is the second most frequently isolated yeast in traditional sourdough fermentation [19].

The acid-producing capacity of 16 LAB strains was investigated by measuring the change in fermentation broth pH. Organic acids such as lactic acid and acetic acid metabolized by bacterial could reduce pH, and more flavor could be added when sourdough was used as a fermentation agent of whole wheat flour [9]. The pH results of fermentation liquid are shown in Figure 3B. Considering the subsequent fermentation in combination with the same yeast, six LAB of different genera with relatively good acid-producing capacities were selected for subsequent experiments: L-1 (*Lactobacillus johnsonii*), L-2 (*Levilactobacillus brevis*), L-5 (*Pediococcus pentosaceus*), L-12 (*Lacticaseibacillus paracasei*), L-16 (*Lacticaseibacillus casei*), and L-18 (*Pediococcus acidilactici*).

Acidification can increase the solubilization of proteins under acidic conditions because it has a positive net charge and also can affect the viscoelastic behavior of the dough and the activity of cereal and bacterial enzymes [25]. The above four strains of acetic acid bacteria L-10, L-11, L-13, and L-15 were relabeled as A-1, A-2, A-3, and A-4, respectively. The four strains of acetic acid bacteria were all *Acetobacter pasteurii*, but their acid producing abilities were different. A-1 was selected for subsequent experiments (Figure 3C). The most prominent manifestation of acetic acid bacteria is the production of acetic acid, which can reduce the pH and inhibit the growth of fungus. The data showed that acetic acid bacteria were relatively common in starter cultures, but this had been overlooked in past studies [24].

#### 2.3.2. Effects of Different Yeast Combinations on the Quality of Whole Wheat Steamed Bread

Specific volume is an important apparent characteristic index to evaluate the quality of steamed bread and is closely related to CO_2_ production by yeasts [26]. The specific volumes of whole wheat steamed bread made by different yeast fermentation combinations are shown in Figure 4A. The larger the specific volume of whole wheat steamed bread, the better quality characteristics of the product. The specific volumes of whole wheat steamed bread fermented by *Saccharomyces cerevisiae* and *Wicherhamomyces anomalus* alone were significantly higher than that of the control, and the former was superior to the latter, which is consistent with the results of previous studies [26]. However, when *Saccharomyces cerevisiae* and *Wicherhamomyces anomalus* were used in combination, the effect was not as good as *Saccharomyces cerevisiae* fermentation alone, because the resource competition between *Saccharomyces cerevisiae* and *Wicherhamomyces anomalus* may not be conducive to the growth of *Saccharomyces cerevisiae* [19]. When fermented with *Saccharomyces cerevisiae*, the specific volume of whole wheat steamed bread increased by 7.6% when compared with the control.

The textural parameters of the steamed bread were explored, including hardness, cohesiveness, springiness, chewiness, and gumminess (Table 3). Hardness is an important parameter affecting the sensory quality of steamed bread and a main index of aging during storage [27]. The whole wheat steamed bread fermented by Y2 had the lowest hardness (802.3 g) and the highest springiness (0.90) values among all treatment groups, indicating a better product quality. Therefore, Y2 was selected for the following experiments. 

#### 2.3.3. Effects of Different LAB Combinations on the Quality of Whole Wheat Steamed Bread 

*Saccharomyces cerevisiae* (Y2) was mixed with *Lactobacillus johnsonii*, *Levilactobacillus brevis*, *Pediococcus pentosaceus*, *Lacticaseibacillus casei*, *Lacticaseibacillus paracasei*, and *Pediococcus acidilactici* to make whole wheat steamed bread (Figure 4B). When yeast and LAB were mixed, the specific volume of whole wheat steamed bread was higher than that of the control. 

Due to the existence of bran, the air holding capacity of whole wheat steamed bread decreases, and the intensity is weakened, so the specific volume of whole wheat flour products is lower than the ideal level [2,10]. When fermented by *Saccharomyces cerevisiae* and *Lactobacillus johnsonii*, the specific volume of whole wheat steamed bread was the largest, reaching 1.81 cm^3^/g. *Saccharomyces cerevisiae* produced gas during the fermentation process, which increased the volume of the steamed bread, and the presence of *Lactobacillus* can promote *Saccharomyces cerevisiae* metabolism, resulting in a higher production, which can improve dough leavening [24,28]. 

When compared with the whole wheat steamed bread fermented by commercial dry yeast, whole wheat steamed bread fermented by *Saccharomyces cerevisiae* and *Lactobacillus johnsonii* had a higher springiness and lower hardness (Table 4). Adding different LAB had different impacts on the texture of whole wheat steamed bread, which illustrates that LAB and *Saccharomyces cerevisiae* may play a synergistic role together [9,24,28].

### 2.4. Effects of Compound Starter on the Quality of Whole Wheat Steamed Bread

Considering the effect of acetic acid bacteria on flavor, *Saccharomyces cerevisiae*, *Lactobacillus johnsonii*, and *Acetobacter pasteurii* were mixed as a compound starter to produce whole wheat steamed bread, denoted as CS-WB. It can be seen from Figure 5 that the height and volume of CS-WB increased more than others. The height and volume of CS-WB were 8.0 and 14.5% greater than DY-WB, respectively, as shown in Table 5. By adding *Lactobacillus johnsonii* and *Acetobacter pasteurii*, the specific volume of CS-WB reached 1.83 cm^3^/g, 5.4% higher than SY-WB, which may involve a synergy among three microorganisms [28]. Changes in dough rise rates and aromas are largely explained by acetic-acid-producing bacteria, which is an often-overlooked group of sourdough microbes [9,29].

In this study, the quality of whole wheat steamed bread was evaluated by an objective method to measure the texture. The textural results of different samples are shown in Table 5. Compared to DY-WB, the hardness and chewiness of CS-WB decreased by 45.2 and 50.1%, respectively, and the springiness of the CS-WB increased by over 10.8%, which indicates that CS-WB has a softer taste. Similar studies have shown that the effect of leavening agents on the hardness of steamed bread was related to CO_2_ production by yeast. Microbial acidification and the gas inside the dough increased and the gluten became weaker, making the dough softer and the final product less firm [9,28].

### 2.5. Determination of Volatile Compounds of Whole Wheat Steamed Bread

Flavor plays an important role in whole wheat steamed bread quality, and is formed by a number of volatiles, including alcohols, aldehydes, ketones, ethers, acids, esters, hydrocarbons, and aromatics. The volatiles of steamed bread are affected by many factors, such as raw materials, leavening agents, microbial strains, and fermentation conditions [30,31,32,33].

In this study, GC-MS was applied to analyze the volatile compounds. A total of 38 volatile compounds were detected in whole wheat steamed bread with the three treatments (Table 6), comprising 6 alcohols, 7 aldehydes, 2 ketones, 5 acids, 14 esters, 2 hydrocarbons, and 2 other compounds. Moreover, 33 volatile compounds were detected in CS-WB, 14 more than in DY-WB, while 31 volatile compounds were detected in SY-WB.

Alcohols are the hydroxyl compounds produced in microbial metabolism. The contents of alcohols were higher in all samples, especially phenylethanol. Phenylethanol mostly exists in natural products such as apple, strawberry, and honey, and has the fragrance of rose, which has a positive correlation with bread aroma [34]. It is worth noting that the content of phenylethanol in CS-WB was the lowest among the three treatments, comprising only 10.51% of its total MS peak area, which may be because the presence of *Lactobacillus johnsonii* and *Acetobacter pasteurii* enhanced the metabolism of phenylethanol by *Saccharomyces cerevisiae*. The molecule 3-methyl-1-butanol has a positive correlation with aroma, and it is the most-cited compound in bread flavor research, with a high odor activity value, and gives a “balsamic, alcoholic, malty” flavor [34]. Interestingly, 3-methyl-1-butanol and phenylethanol showed the same trend of change, with their content decreasing to 1.58% when fermented by the compound starter. The content of four acids increased after compound starter leavening, especially with the acetic acid, reaching 21.37% of its total MS peak area in CS-WB. Esters, aldehydes, and alkanes increased in whole wheat steamed bread added with LAB and acetic acid bacteria. The increasing content of esters may be caused by the esterification reaction of alcohols and acids during the steaming process [33,34]. Aldehydes are the main products of Maillard reactions, and their content was higher than that of DY-WB and SC-WB. Seven types of substances, including hexanal, heptanal, and nonanal chemicals, were detected in CS-WB. Hexanal is commonly found in steamed bread, and heptanal and nonanal are produced through fermentation and lipid oxidation. The smell of hexanal is soapy, fruity, rose, citrus, and orange [26,35]. Phenylacetaldehyde can be used as a fragrance and has a strong rose fragrance. However, the threshold of hydrocarbons is relatively high, so their contribution to the flavor of bread is small. In addition, the content of 2-Pentylfuran of CS-WB was increased from 0.37 to 1.23% when compared to that of DY-WB, and 2-Pentylfuran has the aroma of mushrooms, lavender, and hay [30,33].

The whole wheat steamed bread had higher volatile compounds when fermented by *Saccharomyces cerevisiae*, *Lactobacillus johnsonii*, and *Acetobacter pasteurii*, which produced some unique volatile compounds, such as acetoin, 3-hydroxy-butanal, butyraldehyde, cuparene, etc. The production of these volatile compounds may be caused by the synergistic effect of *Saccharomyces cerevisiae*, *Lactobacillus johnsonii*, and *Acetobacter pasteurii*, leading to reactions among the metabolites to generate a series of secondary metabolites [2,33].

### 2.6. Sensory Evaluation

The sensory evaluation results are shown in Figure 6. It can be seen from the sensory evaluation radar chart that CS-WB was superior to DY-WB in appearance, color, taste, and flavor. The total score of CS-WB was 88.4 points, which was 18.2% higher than that of DY-WB. This indicates a slight preference for the compound starter-fermented whole wheat flour steamed bread. In terms of flavor, the score of CS-WB was 19.1, while that of DY-WB was only 14.2, which indicates that compound fermentation greatly improved the flavor of the product. This was consistent with the flavor results from the GC-MS determination. It has been reported that LAB and acetic acid bacteria show a greater impact on dough systems by lowering dough pH and promoting the production of more alcohol, esters, aldehydes, and other flavor compounds than in yeast fermentation alone, which is beneficial for whole wheat steamed bread to produce better flavors [12,28,33].

## 3. Materials and Methods

### 3.1. Materials

Five sourdough samples were collected from private households in areas of China where steamed bread is a staple food (Shandong, Henan, and Anhui regions). All of the samples were refrigerated during the transfer to the lab and stored at 4 °C in the lab.

Whole wheat flour was produced by an experimental mill (LRMM-8040-3-D, BUHLER, Uzwil, Switzerland) grinding Yang-Mai 16, which was obtained from the Jindadi Seed Industry Co., Ltd. (Suzhou, Jiangsu, China). Active dry yeast (Angel brand, Huanggang, Hubei, China) and sucrose (Ganjuyuan brand, Jiangsu, China) were purchased from a local supermarket. A Bacterial Genomic DNA Extraction Kit and Yeast Genomic DNA Rapid Extraction Kit were purchased from Sangong Bioengineering Co., Ltd. (Shanghai, China). YPD and MRS culture media were obtained from Beijing Aoboxing Co., Ltd. (Beijing, China). All used reagents were of analytical grade unless otherwise specified.

### 3.2. Determination of pH and Total Titratable Acidity

The pH and total titrated acidity (TTA) determination followed the method of De Vuyst et al. [11], with minor modifications. We mixed 10 g sourdough with 90 mL sterile distilled water and homogenized the mixture for 2 min with a magnetic stirrer (FS-2, Changzhou Guohua Electric Co. Ltd., Jiangsu, China). The pH was recorded by a pH meter (HZP-T502, Huazhi Saixijie Technology Co. Ltd., Jiangsu, China). TTA was expressed as the volume of 0.1 mol/L. NaOH was added to neutralize the solution to pH 8.5 in flour.

### 3.3. Counts of LAB and Yeast

Yeast and LAB were counted by the plate-counting method [36]. Yeast was cultured in YPD medium and LAB was cultured in MRS medium. Sourdough (10 g) was suspended in 90 mL sterile physiological saline (0.85%, *w*/*v*). The suspension was diluted to 10^9^ with a 10-fold gradient. Then, 100 μL of 10^4^–10^6^ diluents were plated onto YPD medium and incubated at 28 °C for 36–48 h in an incubator (DRP-9162, Shanghai Senxin Experimental Instrument Co., Ltd., Shanghai, China), and 100 μL of 10^7^–10^9^ diluents were plated onto MRS medium and incubated at 37 °C for 48–72 h.

### 3.4. Isolation and Identification of LAB and Yeast

Sourdough (1 g) was placed in YPD medium and incubated overnight at 28 °C. Taking the proper amount of culture medium, we diluted it with sterile normal saline, removed 100 μL and spread it on YPD solid medium, and then incubated the plates at 28 °C for 30 h [6]. Suspected yeast colonies were selected from the plate by observing their form under electron microscopy (Nikon Eclipse E100, Japan). Finally, single colonies were selected and streaked on YPD solid medium for purification. After repeating this step 3–5 times, pure yeast culture was obtained. The culture medium of LAB was MRS medium, and the incubation time was 48 h. The other steps were the same as the separation and purification of yeast.

Yeast and LAB strains were inoculated with YPD and MRS liquid medium incubated at 28 and 37 °C for the logarithmic growth stage, respectively, and their DNA was extracted by a Genomic DNA Extraction Kit and Genomic DNA Rapid Extraction Kit (Yeast). The DNA concentration and purity were measured by the NanoDrop 2000 UV-vis spectrophotometer (Thermo Scientific, Wilmington, NC, USA), and DNA quality was detected by 1% agarose gel electrophoresis [37]. The characteristics of the fragment of the bacterial 16S rRNA gene were amplified with the primers 1492R: 5′-TACCTTGTTACGACTT-3′ and 27F: S’-AGAGTTTGATCCTGGCTCAG-3′ by using the thermocycler PCR system (PTC-100TM, MJ Research, USA). The characteristics of the fragment of the yeast ITS gene were amplified with the primers ITS1: 5’-TCCGTAGGTGAACCTGCGG-3’ and ITS4:5’-TCCTCCGCTTATTGATATGC-3’. The PCR program was as follows [6]: degeneration at 95 °C for 5 min; 35 cycles of 95 °C for 30 s, 55 °C for 30 s, and elongation at 72 °C for 60 s; a final extension at 72 °C for 10 min. The total reaction volume was a 50 μL mixture containing 3 μL of buffer, 2 μL of 2.5 mM dNTPs, 3 μL of upstream and downstream primers, 0.2 μL Taq enzyme, 1 μL DNA template, and 17.8 μL ddH_2_O.

The bacterial solution obtained from PCR amplification fragment cloning was sent to Jinweizhi Biotechnology Co., LTD, Suzhou, China for sequencing. The sequencing results were compared with the BLAST database in NCBI. According to homologous sequence search and sequence alignment results, MEGA6.05 was used for sequence alignment, and phylogenetic trees were constructed to determine the specific information of dominant species [38].

### 3.5. Screening of the Compound Starter

#### 3.5.1. Comparison of Fermentability of Yeast

The fermentation capacity of yeast was measured by the drainage method [39]. For a reserve, 100 g whole wheat flour was measured and kept at 30 °C. Then, the appropriate amount of sucrose was added into yeast mud and activated for 20 min. We mixed NaCl with yeast mud and whole wheat flour, immediately put it into bottle A with a drainage device, connected bottle B, and checked it for air leakage. We recorded the displacement every 1.0 h.

#### 3.5.2. Comparison of Acid-Producing Capacity of LAB and Acetobacter

LAB and *Acetobacter* were screened by comparing their acid-producing capacities. The activated LAB seed liquor was inoculated in MRS liquid medium with 2% inoculation and incubated at 37 °C. Samples were taken at a certain time, and the pH of fermentation broth was measured. The dominant strains were screened by comparing the pH changes. The comparison method of acid production capacity of *Acetobacter* was the same as that of LAB.

#### 3.5.3. Strain Combination Test of Compound Starter

Three kinds of yeasts were fermented separately and then compounded in pairs (Table 7). Then, the selected optimal yeast was mixed with 6 kinds of LAB to ferment whole wheat steamed bread according to different combinations. The combinations are listed below. Finally, the best combination of yeast and LAB and the dominant *Acetobacter* were selected to form a compound starter.

### 3.6. Whole Wheat Wteamed Bread Making

The whole wheat steamed bread was fermented by commercial dry yeast (DY-WB) according to the following methods [7]. A total of 100 g wheat flour, 1 g Angel yeast, and 70 g warm water (30 °C) were mixed at a low speed (100 rpm) in a needle mixer JHMZ (Beijing Oriental Fude Technology Co., LTD, Beijing, China) for 10 min. The dough was leavened for the first fermentation in a fermenting box (35 °C, 85% relative humidity) for 1 h. After that, we took out the dough, rolled it 10 times with a tablet press, divided the dough into 50 g pieces, kneaded the dough round by hand, put it in a fermenting box at 38 °C, and left it for 30 min for the second fermentation. We then put it into a pot with boiling water and steamed it for 20 min. For the making of whole wheat steamed bread fermented by a single yeast, 100 g whole wheat flour was added with yeast solution at the rate of 3%, and the yeast concentration was adjusted to a uniform concentration of 1.0 × 10^6^ CFU/mL. The remaining steps were the same as commercial dry yeast. For the making of whole wheat steamed bread fermented by compound starter (CS-WB), 100 g of whole wheat flour was added with 3% compound starter (yeast: *Lactobacillus*: acetic acid bacteria = 2:1:1), and the concentration of strains was adjusted (yeast: 1.0 × 10^6^ CFU/mL, *Lactobacillus*: 5.0 × 10^7^ CFU/mL, acetic acid bacteria: 5.0 × 10^7^ CFU/mL). The remaining steps were the same as commercial dry yeast.

### 3.7. Specific Volume, Sensory Evaluation, and Texture Profile Analysis of Whole Wheat Steamed Bread 

The specific volume of whole wheat steamed bread was determined by the rapeseed displacement method (GB/T 21118-2007). Sensory evaluation was carried out by a sensory evaluation group, which consisted of 15 trained members (8 women and 7 men, aged 23–35). The sensory evaluation provides a scale of appearance (20%), color (20%), taste (20%), flavor (20%), and overall acceptability (20%). The texture profile analysis (TPA) was carried out using a TA-XT2i Texture Analyzer (TLP, Food Technology Corporation, Sterling, VA, USA). Thin slices (20 mm) taken from the central part of whole wheat steamed bread were evaluated with a 50 mm diameter cylinder probe [7]. The pre-test speed was 1.0 mm/s, test speed was 1.0 mm/s, post-test speed was 10.0 mm/s, distance was 40%, and the test gap was 30 s. Each sample was taken in parallel 3 times. 

### 3.8. Determination of Volatile Compounds of Whole Wheat Steamed Bread

The volatile compounds in whole wheat steamed bread were detected using solid-phase micro-extraction (SPME) and gas chromatography/mass spectroscopy (GC/MS) according to Kim et al. [40]. The processed whole wheat steamed bread core was divided into fragments (5 mm × 5 mm). We put samples (2 g) into a vial, and sampled them with a 65 μm CAR/DVB/PDMS SPME fiber (Sigma, Saint Louis, MO, USA) at 60 °C for 40 min. The SPME fiber was desorbed for 2 min in the injector port (250 °C) of the GC/MS (7890GC-5975MSD, Agilent, Palo Alto, CA, USA). Separation was achieved on a polar HP-5 capillary column. The GC temperature was at 40 °C for 1 min, increased to 160 °C at a rate of 6 °C/min, and then to 250 °C at a rate of 10 °C/min. Mass spectra were recorded by electronic impact (EI) at 70 eV using ion source temperatures at 200 °C. The scan mode was used to detect all the compounds in the range *m*/*z* 33–495 atomic mass unit (amu). The identification of volatile compounds was performed by matching the obtained mass spectra with those stored in the National Institute of Standards and Technologies (NIST) US Government library (https://webbook.nist.gov, accessed on 10 January 2021).

### 3.9. Statistical Analysis

All the experiments were individually carried out in triplicate; the average and standard deviation values are expressed. Statistical analysis was performed using the SPSS Statistics Software (SAS Institute Inc., 1990, Cary, NC, USA). Comparisons were carried out by the Duncan’s multiple comparison method, and differences were considered statistically significant at *p* < 0.05.

## 4. Conclusions

In this study, three yeasts, six LAB, and one *Acetobacter* were isolated from five different types of traditional Chinese sourdough. Based on the fermentability and acid production capacity, the combination of *Saccharomyces cerevisiae*, *Lactobacillus johnsonii*, and *Acetobacter pasteurianum* in the composite starter was optimized. The result showed that the specific volume of CS-WB was significantly increased, while hardness and chewiness were decreased. In addition, the sensory evaluation score of CS-WB was the highest among three groups. The addition of LAB and acetic acid bacteria enriched the volatile flavor in whole wheat steamed bread, and several unique flavor substances in whole wheat steamed bread were formed after fermentation by the compound starter. Overall, the compound starters comprising *Acetobacter pasteurianum*, *Saccharomyces cerevisiae*, and *Lactobacillus johnsonii* provide a new tool for improving the edible quality of whole wheat fermented flour products.

## Figures and Tables

**Figure 1 molecules-27-03510-f001:**
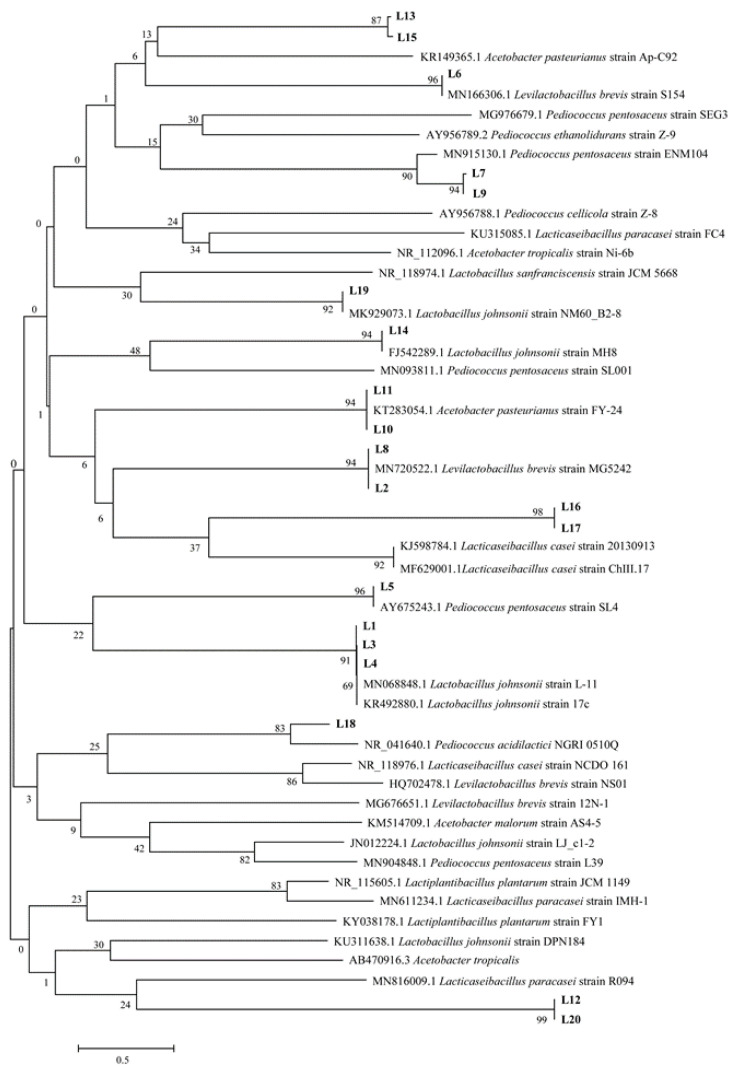
Phylogenetic tree of bacteria based on 16S rDNA gene sequence.

**Figure 2 molecules-27-03510-f002:**
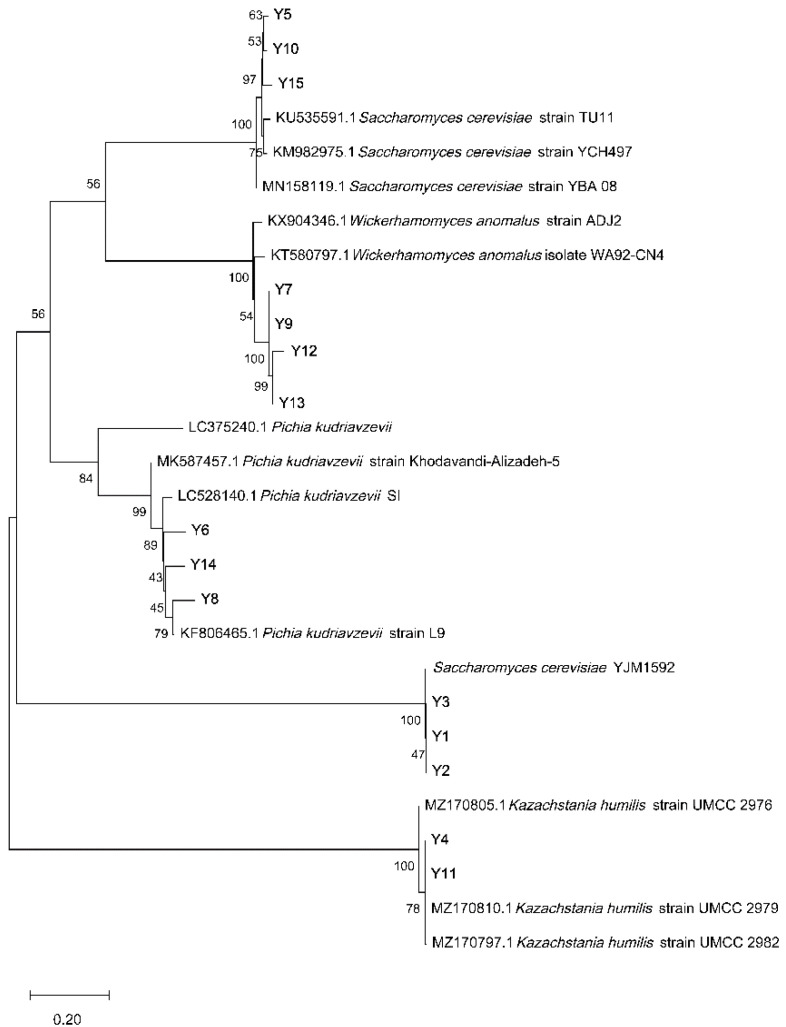
Phylogenetic tree of yeast based on ITS gene sequence.

**Figure 3 molecules-27-03510-f003:**
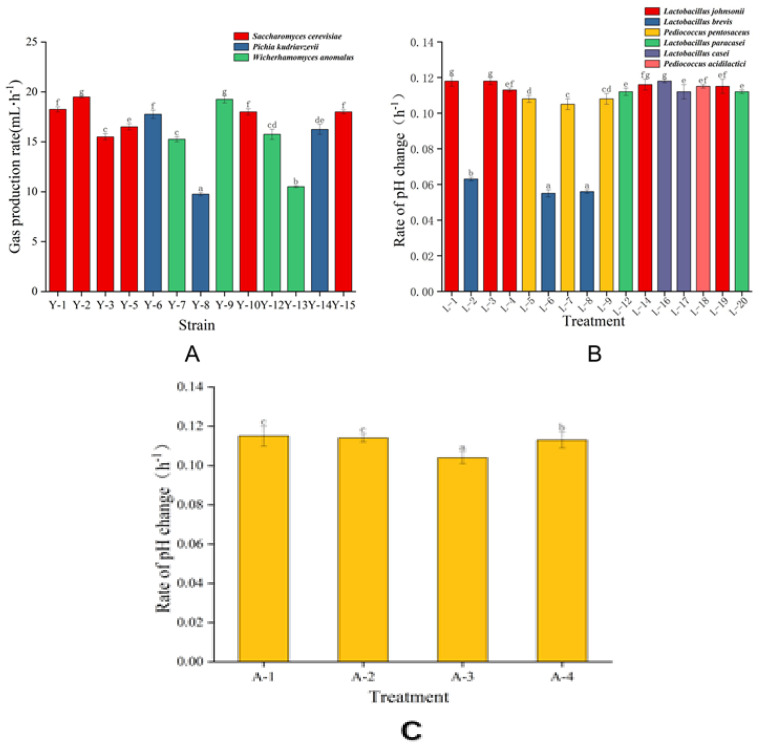
(**A**) Gas production rate of yeasts (mL·h^−1^). (**B**) Rate of fermentation broth pH change of LAB (h^−1^). (**C**) Rate of fermentation broth pH change of acetic acid bacteria (h^−1^). Different superscript letters (a–g) above bars indicate significant difference (*p* < 0.05).

**Figure 4 molecules-27-03510-f004:**
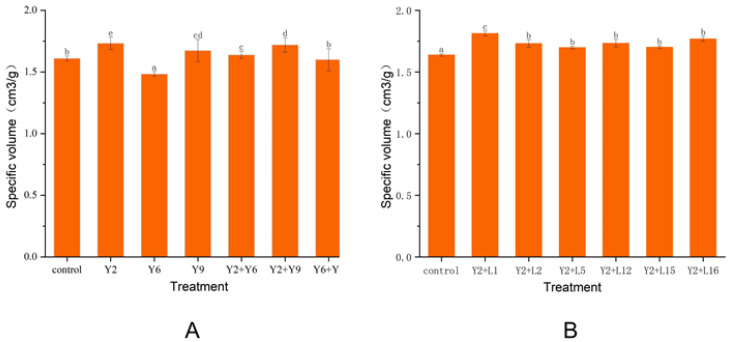
(**A**) Effects of different yeast combinations on the quality of whole wheat steamed bread. (**B**) Effects of different LAB combinations on the quality of whole wheat steamed bread. Note: control—commercial dry yeast; Y2—*Saccharomyces cerevisiae*; Y6—*Pichia kudriavzevii*; Y9—*Wicherhamomyces anomalus*; L1—*Lactobacillus johnsonii*; L2—*Levilactobacillus brevis*; L5—*Pediococcus pentosaceus*; L12—*Lacticaseibacillus paracasei*; L16—*Lacticaseibacillus casei*; L18—*Pediococcus acidilactici*. Different superscript letters (a–e) above bars indicate significant difference (*p* < 0.05).

**Figure 5 molecules-27-03510-f005:**
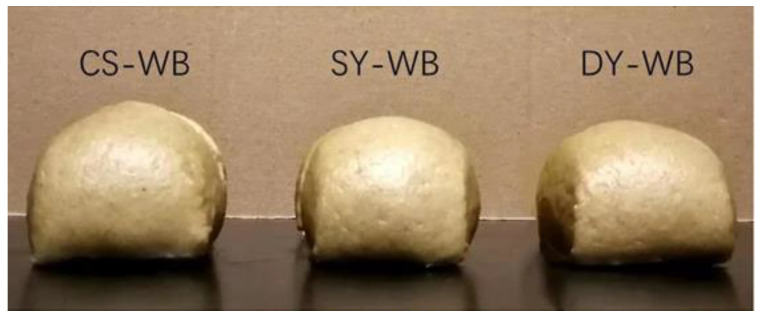
Photographs of whole wheat steamed bread. Note: CS-WB—whole wheat steamed bread fermented by compound starter; SY-WB—whole wheat steamed bread fermented by *Saccharomyces cerevisiae*; DY-WB—whole wheat steamed bread fermented by commercial dry yeast.

**Figure 6 molecules-27-03510-f006:**
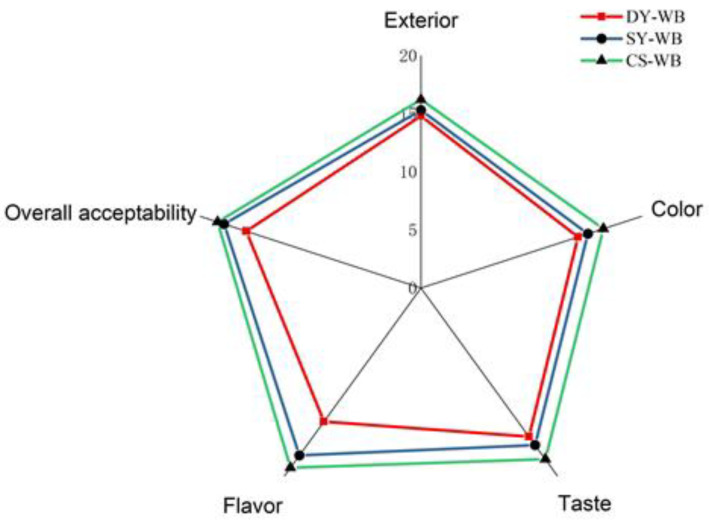
Radar image of sensory evaluation of different whole wheat steamed breads. Note: DY-WB—whole wheat steamed bread fermented by commercial dry yeast; SY-WB—whole wheat steamed bread fermented by *Saccharomyces cerevisiae*; CS-WB—whole wheat steamed bread fermented by compound strain.

**Table 1 molecules-27-03510-t001:** Acidity and microbial counts of CTS.

Sample	Source	pH	TTA (mL)	Microbial Counts (Log cfu/g)	
				Yeast	LAB
CTS 1	Henan	4.56 ± 0.02 ^c^	9.15 ± 0.26 ^c^	6.73 ± 0.06 ^d^	4.96 ± 0.13 ^a^
CTS 2	Henan	5.19 ± 0.02 ^a^	6.40 ± 0.35 ^a^	3.64 ± 0.12 ^a^	6.02 ± 0.08 ^b^
CTS 3	Anhui	3.87 ± 0.04 ^d^	15.27 ± 0.37 ^d^	5.21 ± 0.09 ^b^	7.58 ± 0.15 ^c^
CTS 4	Anhui	3.61 ± 0.06 ^e^	16.32 ± 0.31 ^e^	6.20 ± 0.25 ^c^	9.57 ± 0.21 ^e^
CTS 5	Shandong	5.07 ± 0.05 ^b^	7.28 ± 0.60 ^b^	6.18 ± 0.17 ^c^	8.29 ± 0.11 ^d^

Means in the same column with different superscript lowercase letters of the same modulus indicate significant difference (*p* < 0.05).

**Table 2 molecules-27-03510-t002:** Identification results of yeasts and LAB.

Category	Strain	Results	Homology	Serial Number
LAB	L-1	*Lactobacillus johnsonii*	100	L-11
	L-2	*Levilactobacillus brevis*	99	MG5242
	L-3	*Lactobacillus johnsonii*	99	L-11
	L-4	*Lactobacillus johnsonii*	100	L-11
	L-5	*Pediococcus pentosaceus*	100	SL4
	L-6	*Levilactobacillus brevis*	99	S154
	L-7	*Pediococcus pentosaceus*	100	ENM104
	L-8	*Levilactobacillus brevis*	99	MG5242
	L-9	*Pediococcus pentosaceus*	99	ENM104
	L-10	*Acetobacter pasteurianus*	99	FY-24
	L-11	*Acetobacter pasteurianus*	99	FY-24
	L-12	*Lacticaseibacillus paracasei*	100	R094
	L-13	*Acetobacter pasteurianus*	99	Ap-C92
	L-14	*Lactobacillus johnsonii*	99	MH8
	L-15	*Acetobacter pasteurianus*	98	Ap-C92
	L-16	*Lacticaseibacillus casei*	98	20130913
	L-17	*Lacticaseibacillus casei*	97	20130913
	L-18	*Pediococcus acidilactici*	98	NGRI0510Q
	L-19	*Lactobacillus johnsonii*	99	NM60-B2-8
	L-20	*Lacticaseibacillus paracasei*	98	R094
Yeast	Y-1	*Saccharomyces cerevisiae*	100	YJM1592
	Y-2	*Saccharomyces cerevisiae*	100	YJM1592
	Y-3	*Saccharomyces cerevisiae*	99	YJM1592
	Y-4	*Kazachstania humilis*	96	UMCC 2979
	Y-5	*Saccharomyces cerevisiae*	100	TU11
	Y-6	*Pichia kudriavzevii*	100	SI
	Y-7	*Wicherhamomyces anomalus*	99	WA92-CN4
	Y-8	*Pichia kudriavzevii*	99	L9
	Y-9	*Wicherhamomyces anomalus*	100	WA92-CN4
	Y-10	*Saccharomyces cerevisiae*	99	TU11
	Y-11	*Kazachstania humilis*	95	UMCC 2979
	Y-12	*Wicherhamomyces anomalus*	99	WA92-CN4
	Y-13	*Wicherhamomyces anomalus*	100	WA92-CN4
	Y-14	*Pichia kudriavzevii*	99	SI
	Y-15	*Saccharomyces cerevisiae*	98	TU11

**Table 3 molecules-27-03510-t003:** The effects of different yeast on the texture properties of whole wheat steamed bread.

Treatment	Hardness	Gumminess	Chewiness	Springiness	Cohesiveness
control	1001.9 ± 38.0 ^b^	826.3 ± 15.0 ^c^	711.3 ± 26.8 ^b^	0.86 ± 0.01 ^b^	0.82 ± 0.01 ^a^
Y2	802.3 ± 36.4 ^a^	667.2 ± 15.8 ^a^	600.6 ± 20.9 ^a^	0.90 ± 0.01 ^c^	0.83 ± 0.00 ^b^
Y6	1643.9 ± 36.4 ^d^	1393.8 ± 15.8 ^e^	1084.2 ± 20.9 ^d^	0.81 ± 0.01 ^a^	0.81 ± 0.00 ^b^
Y9	864.1 ± 36.4 ^a^	712.8 ± 15.8 ^b^	626.7 ± 20.9 ^a^	0.88 ± 0.02 ^bc^	0.83 ± 0.00 ^b^
Y2 + Y6	1138.5 ± 36.4 ^c^	934.9 ± 14.8 ^d^	820.5 ± 20.9 ^c^	0.88 ± 0.01 ^bc^	0.82 ± 0.00 ^b^
Y2 + Y9	805.6 ± 35.4 ^a^	667.2 ± 15.8 ^a^	600.6 ± 20.9 ^a^	0.90 ± 0.01 ^c^	0.83 ± 0.00 ^b^
Y6 + Y9	1006.9 ± 31.4 ^b^	829.3 ± 12.8 ^c^	721.3 ± 18.9 ^b^	0.86 ± 0.01 ^b^	0.82 ± 0.00 ^b^

Note: control—commercial dry yeast; Y2—*Saccharomyces cerevisiae*; Y6—*Pichia kudriavzevii*; Y9—*Wicherhamomyces anomalus*. Means in the same column with different superscript lowercase letter of the same modulus indicate a significant difference (*p* < 0.05).

**Table 4 molecules-27-03510-t004:** The effects of different LAB on the texture properties of whole wheat steamed bread.

Treatment	Hardness	Gumminess	Chewiness	Springiness	Cohesiveness
control	1001.9 ± 35.4 ^e^	826.3 ± 15.8 ^e^	711.3 ± 20.9 ^e^	0.84 ± 0.01 ^b^	0.81 ± 0.00 ^a^
Y2 + L1	661.4 ± 31.4 ^a^	542.3 ± 12.8 ^a^	463.7 ± 18.9 ^a^	0.90 ± 0.01 ^e^	0.83 ± 0.00 ^bc^
Y2 + L2	802.3 ± 27.4 ^cd^	667.2 ± 12.0 ^b^	600.6 ± 18 ^cd^	0.80 ± 0.01 ^a^	0.83 ± 0.00 ^c^
Y2 + L5	743.9 ± 21.4 ^b^	693.8 ± 12.0 ^c^	584.2 ± 16.0 ^c^	0.81 ± 0.01 ^a^	0.81 ± 0.00 ^a^
Y2 + L12	764.1 ± 15.4 ^bc^	712.8 ± 12.0 ^cd^	626.7 ± 17.0 ^d^	0.88 ± 0.01 ^d^	0.83 ± 0.00 ^bc^
Y2 + L16	805.6 ± 24.4 ^d^	667.2 ± 14.8 ^b^	530.6 ± 16.0 ^b^	0.86 ± 0.01 ^c^	0.83 ± 0.00 ^c^
Y2 + L18	838.5 ± 15.4 ^d^	734.9 ± 12.8 ^d^	520.5 ± 16.0 ^b^	0.88 ± 0.01 ^d^	0.82 ± 0.00 ^b^

Note: control—commercial dry yeast; Y2—Saccharomyces cerevisiae; L1—Lactobacillus johnsonii; L2—Levilactobacillus brevis; L5—Pediococcus pentosaceus; L12—Lacticaseibacillus paracasei; L16—Lacticaseibacillus casei; L18—Pediococcus acidilactici. Means in the same column with different superscript lowercase letters of the same modulus indicate a significant difference (*p* < 0.05).

**Table 5 molecules-27-03510-t005:** Textural properties of whole wheat steamed bread.

Treatment	Height (cm)	Volume (cm^3^)	Specific Volume (cm^3^/g)	Hardness (g)	Chewiness	Springiness
DY-WB	4.03 ± 0.09 ^a^	80.35 ± 2.07 ^a^	1.62 ± 0.02 ^a^	901.91 ± 18.2 ^c^	700.34 ± 20.3 ^c^	0.83 ± 0.008 ^a^
SY-WB	4.22 ± 0.12 ^b^	85.42 ± 1.50 ^b^	1.73 ± 0.02 ^b^	716.05 ± 16.1 ^b^	585.21 ± 18.9 ^b^	0.88 ± 0.008 ^b^
CS-WB	4.35 ± 0.08 ^b^	92.01 ± 1.71 ^c^	1.83 ± 0.03 ^c^	621.23 ± 15.1 ^ca^	463.95 ± 19.5 ^a^	0.92 ± 0006 ^c^

Note: DY-WB—whole wheat steamed bread fermented by commercial dry yeast; SY-WB—whole wheat steamed bread fermented by *Saccharomyces cerevisiae*; CS-WB—whole wheat steamed bread fermented by compound starter. Means in the same column with different superscript lowercase letters of the same modulus indicate a significant difference (*p* < 0.05).

**Table 6 molecules-27-03510-t006:** Composition of flavor substance in different whole wheat steamed breads.

Category	Compound Name	DY-WB	SY-WB	CS-WB
		Content (%)	Content (%)	Content (%)
Hydrocarbon	Dodecane	/	/	1.53
	Tetradecane	/	0.46	0.49
Alcohols	Cuparene	/	/	0.20
	Ethanol	2.32	3.94	4.69
	2-methylpropanol	/	0.50	0.35
	3-methyl-1-butanol	3.44	3.60	1.58
	1-Pentanol	6.65	7.67	5.97
	Phenylethanol	43.69	18.11	10.51
Esters	Ethyl Hexanoate	/	0.25	/
	Ethyl caprylate	/	4.86	5.97
	Phenethyl formate	/	1.22	/
	Ethyl nonanoate	/	0.50	0.74
	Ethyl caprate	2.76	3.48	5.38
	Ethyl 5-methylnonanoate	1.57	4.19	4.05
	(Z)-Ethyl cinnamate	/	1.68	1.78
	Sulfurous acid, butyl dodecyl ester	/	/	0.35
	Sulfurous acid, pentadecyl pentyl ester	/	/	0.30
	Sulfurous acid, octadecyl pentyl ester	/	0.38	/
	Methyl myristate	/	3.23	/
	Diisobutyl phthalate	0.52	4.02	0.94
	Methyl 14-methylpentadecanoate	7.32	6.29	5.08
	Octyl P-methoxycinnamate	1.79	1.93	1.92
Acid	Acetic acid	8.29	11.65	21.37
	Propionic acid	7.54	5.83	7.01
	Octanoic acid	0.22	0.42	1.28
	Isovaleric acid	0.37	1.97	2.52
	Hexanoic acid	1.34	1.22	1.58
Aldehydes	Hexanal	0.15	0.42	0.54
	3-hydroxy-Butanal	/	/	0.30
	Heptanal	0.075	0.21	0.64
	Octanal	/	1.42	1.43
	Phenylacetaldehyde	/	0.084	0.35
	Nonanal	/	0.46	0.39
	Butyraldehyde	/	/	0.59
Ketones	Acetoin	/	/	0.64
	2-Nonadecanone	2.39	2.35	/
Other	2-Pentylfuran	0.37	1.22	1.23
	Indole	9.19	6.45	8.29

Note: DY-WB—whole wheat steamed bread fermented by commercial dry yeast; SY-WB—whole wheat steamed bread fermented by *Saccharomyces cerevisiae*; CS-WB—whole wheat steamed bread fermented by compound starter. Content: The content of the volatile compounds is expressed as relative MS peak areas (MS peak area of each compound/total MS peak area × 100).

**Table 7 molecules-27-03510-t007:** Combinations of yeast and LAB.

NO.	Combinations of Yeast	Combinations of Yeast and LAB
1	*Saccharomyces cerevisiae*	*Saccharomyces cerevisiae + Lactobacillus johnsonii*
2	*Pichia kudriavzevii*	*Saccharomyces cerevisiae + Levilactobacillus brevis*
3	*Wicherhamomyces anomalus*	*Saccharomyces cerevisiae + Pediococcus pentosaceus*
4	*Saccharomyces cerevisiae +* *Pichia kudriavzevii*	*Saccharomyces cerevisiae + Lacticaseibacillus paracasei*
5	*Saccharomyces cerevisiae + Wicherhamomyces anomalus*	*Saccharomyces cerevisiae+ Lacticaseibacillus casei*
6	*Pichia kudriavzevii +* *Wicherhamomyces anomalus*	*Saccharomyces cerevisiae + Pediococcus acidilactici*

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
