# Peer review of "Screening of Sourdough Starter Strains and Improvements in the Quality of Whole Wheat Steamed Bread"

_molecules, 2022, doi:10.3390/molecules27113510_

Round 1

Reviewer 1 Report

In this study, yeast, lactic acid bacteria, and acetic acid bacteria were isolated from Chinese traditional sourdough to enhance the organoleptic quality of whole wheat steamed bread. Overall, this manuscript is written well and has merits for publication. Nonetheless, some comments as bellow should be considered first. Abbreviations CS-WB and DY-WB should be explained in the abstract. Also, please explain the novelties more clearly as the use of isolated strains for improving bread quality has already been examined. Manuscript van benefit from language editing. In refence No. 14, the name of the journal is incomplete. Please correct and also check for other references. Reference No. 33 starts with capital letter for the paper name for all vocabularies which is different from other references, please carefully check others and correct.

Author Response

Response to Reviewer 1 Comments

Comments and Suggestions: In this study, yeast, lactic acid bacteria, and acetic acid bacteria were isolated from Chinese traditional sourdough to enhance the organoleptic quality of whole wheat steamed bread. Overall, this manuscript is written well and has merits for publication. Nonetheless, some comments as bellow should be considered first. Abbreviations CS-WB and DY-WB should be explained in the abstract. Also, please explain the novelties more clearly as the use of isolated strains for improving bread quality has already been examined. Manuscript van benefit from language editing. In refernce No. 14, the name of the journal is incomplete. Please correct and also check for other references. Reference No. 33 starts with capital letter for the paper name for all vocabularies which is different from other references, please carefully check others and correct.

Response: Dear Reviewer: We would like to thank you for your comments and providing good suggestions for my manuscript. We have revised my manuscript according to your suggestion. My explanation to the comments point- by- point is as follows.

Point 1: Nonetheless, some comments as bellow should be considered first. Abbreviations CS-WB and DY-WB should be explained in the abstract.

Response 1: We have been added the explanation for abbreviations CS-WB and DY-WB in the abstract, please see in "Line 17-18" for details.

Point 2: Also, please explain the novelties more clearly as the use of isolated strains for improving bread quality has already been examined.

Response 2: Thank you for your precious revision suggestions. Although Saccharomyces cerevisiae, Lactobacillus johnsonii and Acetobacter pasteurii had been isolated from sourdoughs, but there is no combination of them in whole wheat steamed bread making so far. In our study, Saccharomyces cerevisiae, Lactobacillus johnsonii and Acetobacter pasteurii were mixed to ferment the whole wheat steamed bread for the first time, and a comprehensive investigation of whole wheat steamed bread were carried out, including its texture, specific volume, volatile compounds and sensory evaluation. We have also revised in Line73-79 to make it more clear.

Point 3: Manuscript van benefit from language editing.

Response 3: We have revised the language of the entire article through the English edited service of the editorial department of Molecular.

Point 4: In reference No. 14, the name of the journal is incomplete. Please correct and also check for other references. Reference No. 33 starts with capital letter for the paper name for all vocabularies which is different from other references, please carefully check others and correct.

Response 4: We have been completed the name of the journal in the references section including but not limited to reference No. 14. Furthermore, we have checked all references and unify the format of these references that meets the requirement of the journal, especially capitalizing reference titles including but not limited to reference No. 33.

Reviewer 2 Report

The manuscript presents interesting results unfortunately it has several flaws.

The writing needs a thoroughly revision of English language and generally denotes poor care.

Some issues there is a confusion between species and strains

Yeast and bacteria (line 115-116)

Genera names with without capital letter (acetobacter, lactobacillus)

johnsonii spelled incorrectly several times.

Some former Lactobacillus species (for instance L. casei or paracasei) were re-classified. The authors should be aware of that.

Abbreviations that are used without a first explanation (namely in the abstract but also in the results)

Some data that are presented does not offer any information (for instance integral area in table 6)

No explanation on the criteria used to choose the strains. Was one of each genus?

Author Response

Response to Reviewer 2 Comments

Comments and Suggestions: The manuscript presents interesting results unfortunately it has several flaws. The writing needs a thoroughly revision of English language and generally denotes poor care. Some issues there is a confusion between species and strains, Yeast and bacteria (line 115-116).Genera names with without capital letter (acetobacter, lactobacillus) and johnsonii spelled incorrectly several times. Some former Lactobacillus species (for instance L. casei or paracasei) were re-classified. The authors should be aware of that. Abbreviations that are used without a first explanation (namely in the abstract but also in the results).Some data that are presented does not offer any information (for instance integral area in table 6).No explanation on the criteria used to choose the strains. Was one of each genus?

Response: Dear reviewer, thank you for reviewing our manuscript and for the constructive comments, which greatly helped us to improve the manuscript. We have heavily revised our experiments. The manuscript was carefully revised and point-by-point response was listed below. We hope that your comments have been addressed accurately.

Point 1: The writing needs a thoroughly revision of English language and generally denotes poor care.

Response 1: We have revised the language of the entire article through the English edited service of the editorial department of Molecular.

Point 2: Some issues there is a confusion between species and strains, Yeast and bacteria (line 115-116).

Response 2: Thank you for your valuable comments, we have corrected the mistake. When we expressed “Saccharomyces cerevisiae is the dominant yeast in sourdough. And Wicherhamomyces anomalus is also a common yeast found in sourdough,......”(line 113-115), we’d like to illustrate that Saccharomyces cerevisiae and Wicherhamomyces anomalus is the concept of species, belonging to the yeast, while a strain must be an individual within a specie. Obviously, we have been realized that Wicherhamomyces anomalus belongs to the yeast,but not the bacteria (line 115).

Point 3: Genera names with without capital letter (acetobacter, lactobacillus) and johnsonii spelled incorrectly several times.

Response 3: We have been capitalized acetobacter and lactobacillus, and had johnsonii spelled correctly (line 120-127, 190-231,379-437).

Point 4: Some former Lactobacillus species (for instance L. casei or paracasei) were re-classified. The authors should be aware of that.

Response 4: Thanks for your previous suggestions, we have corrected the name of Lactobacillus species in entire article.

Point 5: Abbreviations that are used without a first explanation (namely in the abstract but also in the results)

Response 5: We have been added the explanation for abbreviations CS-WB and DY-WB in the abstract and the results, please see in "Line 17-18" for details.

Point 6: Some data that are presented does not offer any information (for instance integral area in table 6)

Response 6: We have deleted the unnecessary information in Table 6 and added note at the bottom of table. In addition, we describe the content of the Table 6 in the paper (Line 267-300).

Point 7: No explanation on the criteria used to choose the strains. Was one of each genus?

Response 7: We selected strains with higher acid production in different genera, so there was one strain per genus. We did this to take into account possible interactions with subsequent combinations with yeast, because Lactic acid bacteria of different genera may have different results when combined with the same yeast (Line 151-153).

Reviewer 3 Report

Authors isolated various starters from different traditional Chinese sourdough and optimized the combination of yeast, lactic acid bacteria and acidic acid bacteria based on their fermentability and acid production capacity. In general, authors did a comprehensive study and collected many interesting data. The following comments should be addressed before its acceptance for publication.

The aims of the study should be presented clearly.

How authors characterized the volatile compounds? Comparing with NIST database? It will be great if authors could provide the linear retention index values of the volatiles in Table 6.

The integral area is MS peak area? Based on my understanding, the MS peak area is not very accurate as FID peak area.

Author Response

Response to Reviewer 3 Comments

Comments and Suggestions: Authors isolated various starters from different traditional Chinese sourdough and optimized the combination of yeast, lactic acid bacteria and acidic acid bacteria based on their fermentability and acid production capacity. In general, authors did a comprehensive study and collected many interesting data. The following comments should be addressed before its acceptance for publication.

The aims of the study should be presented clearly.

How authors characterized the volatile compounds? Comparing with NIST database? It will be great if authors could provide the linear retention index values of the volatiles in Table 6.

The integral area is MS peak area? Based on my understanding, the MS peak area is not very accurate as FID peak area.

Response : Dear Reviewer, Thank you for your kind letter and your careful work regarding our manuscript. We have revised the manuscript in accordance with your comments. And point-by-point responses to the comments were as follows:

Point 1: The aims of the study should be presented clearly.

Response 1: We have revised in Line 73-79.

Point 2: How authors characterized the volatile compounds? Comparing with NIST database? It will be great if authors could provide the linear retention index values of the volatiles in Table 6.

Response 2: We characterize the volatile compounds by two aspects. Firstly, the identification of the volatile compounds was performed by comparing the obtained mass spectra with those stored in the National Institute of Standards and Technologies (NIST) US Government library. Secondly, the volatile components were relatively quantified by its peak area of mass spectroscopy total ion chromatograms.

Point 3: The integral area is MS peak area? Based on my understanding, the MS peak area is not very accurate as FID peak area.

Response 3: The integral area is the peak area of mass spectroscopy total ion chromatograms. In order to further show the difference and change of the relative content of volatiles between groups, we converted the MS peak area to relative peak area (%). Gas Chromatography-Mass Spectrometer (GC-MS) can be very convenient for the qualitative study of the structure of unknown compounds in mixtures, but it is more difficult and not accurate than flame ionization detector (FID) in quantitative operation since it involves more correction process. We are very sorry that we did not have the FID data due to the limited laboratory condition. As far as we are concerned, there may be a big difference between the GC-MS and GC-FID, and GC-MS is enough to allow us to observe changes in the relative MS peak areas and species of various volatiles between different treatment groups, which is what we want.

Reviewer 4 Report

It is a well-planned and well-written study, congratulations.

In lines 106-107, the authors stated that “By comparing the sequences homology of dominant strains in the gene bank, the sequence homology higher than 98% were selected (Table 2) to build the phylogenetic tree” But there are strains given in table 2 which have lower homology than 98%.

The resolution of fig 2 is not at the desired level.

Line 147, please remove one of the “of” from the sentence.

Line 197-199. How can the authors say that the higher springiness indicates the best taste?

Author Response

Response to Reviewer 4 Comments

Comments and Suggestions: It is a well-planned and well-written study, congratulations.

In lines 106-107, the authors stated that “By comparing the sequences homology of dominant strains in the gene bank, the sequence homology higher than 98% were selected (Table 2) to build the phylogenetic tree” But there are strains given in table 2 which have lower homology than 98%.

The resolution of fig 2 is not at the desired level.

Line 147, please remove one of the “of ” from the sentence.

Line 197-199. How can the authors say that the higher springiness indicates the best taste?

Response: Thanks for making time for reviewing and giving detailed modification comments. We have revised the manuscript accordingly, and present our point-by-point answers to your comments below.

Point 1: In lines 106-107, the authors stated that “By comparing the sequences homology of dominant strains in the gene bank, the sequence homology higher than 98% were selected (Table 2) to build the phylogenetic tree” But there are strains given in table 2 which have lower homology than 98%.

Response 1: Thank you for your precious revision suggestions. We have made the necessary amendments that the sequence homology higher than 95% were selected (Table 2) to build the phylogenetic tree. Since that is enough to explain the affinities between different species, which is exactly what we want. In the study of phylogenetic classification, the phylogenetic trees can visualize the evolutionary relationship between species.

Point 2: The resolution of fig 2 is not at the desired level.

Response 2: We thank you for this suggestion and we had made the suggested changes.

Point 3: Line 147, please remove one of the “of ” from the sentence.

Response 3: Thanks for the valuable comment, we removed the ”of” (Line 149).

Point 4: Line 197-199. How can the authors say that the higher springiness indicates the best taste?

Response 4: Thank you for your precious revision comments. We consulted the literature on this issue and found no connection between the springiness and the taste. Sorry for this mistake we have deleted it (Line 199-208).

Round 2

Reviewer 2 Report

-Please indicate the meaning of the letters in tables 1, 3 and 5

Line 573- 595 - If the acid producing capacity determination was perforemd exactly as the LAB the sections 3.5.2 and 3.5.3 can be fusioned .

Reviewer 3 Report

can be accepted

Author Response

Dear Reviewer :

Thank you for your kind letter and your careful work regarding our manuscript.